# New Cytotoxic Cytochalasans from a Plant-Associated Fungus *Chaetomium globosum* kz-19

**DOI:** 10.3390/md19080438

**Published:** 2021-07-31

**Authors:** Tantan Li, Yun Wang, Li Li, Mengyue Tang, Qinghong Meng, Cun Zhang, Erbing Hua, Yuehu Pei, Yi Sun

**Affiliations:** 1Institute of Chinese Materia Medica, China Academy of Chinese Medical Sciences, Beijing 100700, China; tantanna0309@163.com (T.L.); ywang@icmm.ac.cn (Y.W.); lotusun69@163.com (M.T.); qinghongmeng0813@hotmail.com (Q.M.); czhang@icmm.ac.cn (C.Z.); 2College of Biotechnology, Tianjin University of Science & Technology, Tianjin 300457, China; 3School of Pharmacy, Harbin Medical University, Harbin 150081, China; 4Institute of Materia Medica, Chinese Academy of Medical Sciences & Peking Union Medical College, Beijing 100050, China; annaleelin@imm.ac.cn

**Keywords:** *Chaetomium globosum* kz-19, endophytic fungus, cytochalasans, cytotoxicity

## Abstract

Four new cytochalasans, phychaetoglobins A–D (**1**–**4**), together with twelve known cytochalasans (**5**–**16**), were isolated from a mangrove-associated fungus *Chaetomium globosum* kz-19. The new structures were elucidated on the basis of extensive 1D and 2D NMR, HR ESIMS spectroscopic analyses, and electronic circular dichroism (ECD) calculations. The absolute configuration of **2** was established by application of Mosher’s method. Compounds **4**–**8** exhibited moderate cytotoxicities against A549 and HeLa cell lines with the IC_50_ values less than 20 μM.

## 1. Introduction

Natural products from plant-associated fungi are a rich source of biologically promising lead compounds. The fungus genus *Chaetomium* is the rich source of cytochalasans, among which *Chaetomium globosum* can produce plenty of such secondary metabolites [1,2,3,4,5]. As the main secondary metabolites from *C. globosum*, cytochalasans are a class of compounds containing isoindolinone moieties fused to the tricyclic or tetracyclic ring systems, which have a variety of pharmacological activities, such as antitumor [6], antibacterial [7], immunomodulatory [8], and phytotoxicity [9]. The polyketo–amino acid hybrid structure of cytochalasans forms different fused macrocycles (typical 11- or 13-membered macrocycles) of cytochalasans, such as macrocycle, polycyclic, helix, epoxide, hydrogen peroxide groups and other unique structures. Biogenetically, cytochalasans are biosynthesized through a hybrid pathway of polyketide synthase (PKS) and non-ribosomal peptide synthetase (NRPS). The *che* cluster in *Penicillium expansum* was recognized as the first gene cluster encoding PKS-NRPS in the biosynthesis of cytochalasans [10]. In addition, the oxidative tailoring enzymes of CcsB and CHGG_012343 catalyzed multiple steps involved in the process of biosynthesis [11]. In terms of anti-tumor pharmacological effects, many researchers found that cytochalasans could regulate the morphology and phenotype of tumor cells and induce apoptosis [12]. Thus, cytochalasans have the potential to be used as a supplementary chemotherapeutic drug and to decrease the incidence of drug resistance clinically in the future [13]. 

In our continuous research on antitumor secondary metabolites from endophytic microorganisms [14], we used the bioassay method to screen fungi. During the screening process, we found an endophytic fungus *Chaetomium globosum* kz-19, which was isolated from the mangrove plant, *Ceriops tagal*. Its EtOAc extract of rice cultures exhibited strong cytotoxicity against HeLa cell lines (IC_50_ = 11.0 μg/mL). A large-scale solid culture followed by repeated chromatographic isolation led to the identification of four new cytochalasans phychaetoglobins A–D (**1**–**4**), together with twelve known cytochalasans, chaetoglobosin C (**5**) [15], chaetoglobosin E (**6**) [15], chaetoglobosin G (**7**) [16], chaetoglobosin V (**8**) [17], chaetoglobosin Fex (**9**) [4], isochaetoglobosin D (**10**) [4], chaetoglobosin J (**11**) [15,18], armochaetoglobosin G (**12**) [5], penochalasin G (**13**) [19], penochalasin J (**14**) [20], prochaetoglobosin III_ed_ (**15**) [21], and chaetoglobosin D (**16**) [15,18] (Figure 1).

## 2. Results and Discussion

### 2.1. Structure Elucidation

Compound **1** was obtained as a pale yellow solid. Its molecular formula was deduced as C_32_H_36_N_2_O_5_ by HR ESIMS and NMR data, indicating 16 degrees of unsaturation. Analysis of the ^1^H-NMR spectrum of **1** (Table 1) suggested the presence of two amide protons at *δ*_H_ 11.11 (d, 1.4 Hz) and 8.52 (s), five aromatic protons at *δ*_H_ 7.56 (d, 7.7 Hz), 7.29 (d, 7.9 Hz), 7.26 (d, 2.2 Hz), 7.14 (t, 7.2 Hz) and 7.02 (t, 7.2 Hz), three olefinic protons at *δ*_H_ 5.43 (dd, 15.2, 10.4 Hz), 5.13 (d, 11.1 Hz) and 4.78 (td, 15.0, 3.6 Hz), two methyl singlets (*δ*_H_ 1.23 and 1.29), and two methyl doublets at *δ*_H_ 0.88 (d, 6.5 Hz) and 0.99 (d, 7.4 Hz). Determination of the ^13^C NMR combined with the ^1^H NMR spectra of **1** (Table 2) revealed the presence of three carbonyl groups (*δ*_C_ 204.7, 175.0, and 172.8), a 3-substituted indolyl group (*δ*_C_ 108.0, 111.7, 118.2, 118.9, 121.1, 125.8, 127.8, 136.1), and three oxygenated groups (*δ*_C_ 84.9, 59.9, 56.7), which had the characteristic skeleton of cytochalasans closely resembling chaetoglobosin C (**5**). Further analyses of ^1^H and ^13^C NMR, as well as the 2D NMR spectra of **1** (Figure 2a), supported the presence of an indolyl moiety, a pyrrolidine-2-one, and a cyclohexane ring in the structure, as was the case with that of **5**. However, the differences between **1** and **5** were that **1** possessed two fewer ketone groups and one more ester group than **5**. The HMBC correlations from H-19 to C-17, C-18, and C-20, as well as from H-21 to C-20, C-22, and C-23, revealed the presence of a lactone ring. Further HMBC correlations from H-8 to C-9, C-13, C-14, and C-23, as well as from 18-CH_3_ to C-17, C-18, and C-19, indicated the presence of two olefinic bonds and a ketone group on the macrocycle. The relative configuration of **1** was determined by a NOESY spectrum (Figure 2b) and proton–proton coupling constants. A large coupling constant (*J* = 15.2 Hz) observed between H-13 and H-14 indicated the *E*-geometry of Δ^13^-double bond. Additional coupling constants (*J* = 10.4, 5.2 Hz) of H-8 with H-13 and H-7 in the ^1^H NMR spectrum revealed that H-8 was orientated axial, and the NOESY cross-peaks of H-8 with H-4 and H-5 further confirmed the twist-boat conformation of the cyclohexane ring. The observed NOESY cross-peaks of H-15α with H-13 and H-17, CH_3_-18 with H-16, as well as H-17 with CH_3_-16, implied an α-orientation at CH_3_-16 and the *E*-geometry of Δ^17^-double bond. Consequently, the NOESY cross-peaks of H-3 with CH_3_-11, and CH_3_-12 with H-7, suggested that they were cofacial and in the α-orientation. Additionally, the NOESY correlations of H-22 with H-19 and H-21α, together with the coupling constant (*J* = 8.8 Hz) between H-19 and H-22, revealed the β-orientation of the five-numbered lactone ring. To determine the absolute configuration of **1**, the theoretically calculated electronic circular dichroism (ECD) spectra were performed using time-dependent density functional theory (TDDFT). The Merck Molecular Force Field (MMFF) conformational search in an energy window of 10 kcal/mol, which was optimized at the *ω*B97X/TXVP level (in MeOH), yielded two low-energy geometries with a Boltzmann population of over 1% (Appendix A). The configurations of **1a** and **1b** (**1b** was the enantiomer of **1a**) were compared using the ECD calculation at B3LYP level (Appendix A). The experimental ECD curve of **1** was in agreement with the computed curve of **1a** (Figure 2c). Therefore, the absolute structure of **1** was finally established as 3*S*,4*R*,5*S*,6*R*,7*S*,8*R*,9*S*,16*S*,19*S*,22*R* and named phychaetoglobin A.

Compound **2** was obtained as a colorless solid, and the molecular formula was determined to be C_31_H_38_N_2_O_3_ based on the HRESIMS. A comparison of the ^1^H and ^13^C NMR spectra for **2** (Table 1 and Table 2) with those of penochalasin J (**14**) suggested that **2** had a similar skeleton to **14**. However, the signals for a double bond at C-17/C-18, a methyl at C-18, and a ketone group at C-20 in **14** were missing in the NMR spectra of **2**. The information was supported by the HMBC correlations from CH_3_-16 to C-15, C-16 and C-17, and from H-18 to C-17, C-19, and C-20, as well as by the ^1^H-^1^H COSY spin system from H-16 to H-22. These correlations revealed the presence of three methylenes and a methine connected with a hydroxyl group at C-19. Further HMBC correlations from a pair of olefinic protons, H-21/H-22 to C-20 and C-23, also indicated the presence of an α, β-unsaturated ketone moiety at C-21, C-22, and C-23. The above observations confirmed the planar structure of **2**. The relative configuration of **2** was deduced by a NOESY analysis, which was similar to that of **1**. The NOESY cross-peaks of H-4/H-8/Me-11 inferred that they had the same β-orientations at these positions as **1**. Moreover, Me-16 showed NOESY correlations with H-18 and H-14, which indicated the α-orientation of Me-16. The absolute configuration of C-19 was identified by the modified Mosher’s method (Figure 3). The methyl ester alcohol of **2** was treated with *R*-(−)- and *S*-(+)-α-methyoxy-α-(trifluoromethyl) phenyl acetyl chloride (MTPA-Cl) to afford the *S*- and *R*-MTPA esters (**2****c** and **2****d**), respectively. Analysis of the ^1^H NMR and ^1^H-^1^H COSY spectra led to the assignment of both esters’ chemical shifts in proximity at C-19. The results of Δ*δ_S-R_* values confirmed that the absolute configuration of C-19 was *R* (Figure 3). In addition, the ECD spectrum of **2** was determined and the Cotton effects were identical with the calculated curve of the enantiomer **2a** (Figure 2c and Appendix A). Thus, compound **2** was identified and named phychaetoglobin B (Figure 1).

The molecular formula of **3** was determined to be C_34_H_40_N_2_O_7_ by HRESIMS, requiring 16 degrees of unsaturation (Table 1 and Table 2). The ^13^C NMR and HSQC spectra of **3** showed the presence of three ketones, two amide or ester groups, five methyls, and two oxygenated carbons. Analysis of the ^1^H and ^13^C-NMR data of **3** indicated that it was similar in structure to compound **7**. However, when compared to **7**, the differences were the absence of a pair of double bonds at C-5/C-6, and the presence of a hydroxyl group and a methyl group at C-6 in **3**, which was implied by the HMBC correlations from Me-12 to C-5, C-6, and C-7. Additional HMBC correlations from H-7 to C-8, C-9, C-13, C-24, and C-25, as well as from Me-25 to C-24, revealed the presence of an ethoxycarbonyl group at C-7. The NOESY correlations of H-8 with H-4 and H-5 indicated that they had an α-orientation, whereas the NOE correlations of CH_3_-12 with H-4 and CH_3_-11, as well as CH_3_-11 with H-3, suggested the configuration of the cyclohexane ring. The absolute configuration of **3** was identified by comparing its experimental and calculated ECD data (Appendix A). Thus, **3** was determined as 3*S*, 4*R*, 5*S*, 6*R*, 7*S*, 8*R*, 9*S* and named phychaetoglobin C.

Compound **4** was determined as C_32_H_38_N_2_O_4_ on the basis of HR ESIMS. Its NMR data suggested that **4** was an analogue of **8**. However, **4** lacked the hydroxyl group on the cyclopentenone ring that was observed in **8**. This finding was further confirmed by the HMBC correlations from CH_3_-18 to C-17 and C-19, as well as from H-19 to C-17, C-20, and C-21. The relative configuration of the five-membered ring from C-17 to C-21 was assigned by the NOESY spectrum. The NOE cross-peaks of H-16/H-17 and CH_3_-16/H-21 indicated that the methyl at C-16 and H-21 were in the α-orientation, whereas H-16 and H-17 were at the opposite side. The ECD spectrum was determined to confirm the absolute configuration, which was further compared to the experimental spectrum. The ECD spectrum generated for the cyclopentenone ring was 16*S*, 17*S*, 21*S*, which was consistent with the experimental data of **4a** (Appendix A). Therefore, the absolute configuration was established as in Figure 1 and named phychaetoglobin D.

### 2.2. Biological Assay

Compounds **1**–**16** were evaluated for their cytotoxicity against HeLa human colon adenocarcinoma cell lines and A549 human lung adenocarcinoma cells by MTT methods (Table 3). Adriamycin was used as a positive control. Compounds **5** and **7** exhibited the strongest cytotoxicity against A549 cell lines, with the IC_50_ values below 10 μM. Furthermore, compounds **4**–**8** showed a moderate cytotoxicity against HeLa cell lines, with IC_50_ values of 3.7~10.5 μM, implying that the moieties of a double bond at C-5/C-6 and a hydroxyl group at C-7 or an epoxide ring at C-6/C-7 were the characteristics primarily responsible for the cytotoxicity. Compounds **3**, **10**, **11**, and **13**–**16** exhibited cytotoxic activity (IC_50_ values of 12.2~33.7 μM), indicating that the presence of a ketone group other than a hydroxyl group at C-20 could increase the bioactivity. However, compound **1** displayed weak activity, which suggested that the lactone ring in **1** decreased the cytotoxicity. Compound **2** showed weak cytotoxicities, as it did not possess the functional groups related to the bioactivity. Thus, the five-membered lactone ring between C-17 and C-21 in **1** might decrease the cytotoxicity. 

## 3. Materials and Methods

### 3.1. General Experimental Procedures

Optical rotations were measured with a PerkinElmer 241 polarimeter. NMR spectra were performed on a Bruker ARX-600 spectrometer (600 MHz, Bruker Co., Ltd., Karlsruhe, Germany), and the ^1^H and ^13^C NMR chemical shifts were recorded with the solvent peaks for DMSO-*d*_6_ (*δ*_H_ 2.50 and *δ*_C_ 39.50). High-resolution electrospray ionization mass spectrometry (HRESIMS) data were obtained on a Waters Vion QTOF/MS spectrometer (Waters Mocromass, Manchester, UK) in positive electrospray ionization mode. ECD spectra were recorded on a JASCO J-815 spectrometer (Tokyo, Japan). A UPLC reversed phase C18 analytical column (35 °C, 2.1 mm × 100 mm, 1.7 μm, BEH, Waters) was adopted. High performance liquid chromatography (HPLC) was carried out on a Agilent 1260 quaternary system with a UV detector (Agilent, Technologies Co., Ltd., Palo Alto, CA, USA), combined with analytical, semi-preparative or preparative Cosmosil C_18_-MSII columns (250 mm × 4.6 mm, and 250 mm × 10 mm). Thin-layer chromatography (TLC) was performed with a silica gel plate GF254 (Qingdao Haiyang Chemical Co., Ltd., Qingdao, China). Column chromatography was applied on a Sephadex LH-20 (Pharmacia Fine Chemical Co., Ltd., Uppsala, Sweden), ODS (50 μm, YMC Japan) and silica gel (200–300 mesh, Qingdao Haiyang Chemical Ltd., Qingdao, China). Human carcinoma cell lines HeLa and A549 were obtained from the Chinese National Infrastructure of Cell Line Resource (NICR).

### 3.2. Fungal Material

The endophytic fungal KZ-19 was isolated from twigs of the mangrove plant *Ceriops tagal*, which was collected in Hainan province, China, in July 2013. The plant species was identified by Yi Sun, and the fungus was identified as a *Chaetomium globosum* by its rRNA gene sequence. The strain was deposited at the institute of Chinese Materia Medica, China Academy of Chinese Medical Sciences.

### 3.3. Fermentation and Extraction

According to the previous investigation of culture conditions, the strain adopted the method of solid fermentation. The strain frozen at −80 °C was taken out and cultured on a plate of potato dextrose agar (PDA) medium at 27 ± 0.5 °C for 3 days. The mycelium was inoculated aseptically to 500 mL Erlenmeyer flasks, each containing 40 g of rice and 60 mL of distilled water. There were 100 flasks in total, and the flask cultures were incubated at 27 ± 0.5 °C for 7 days.

### 3.4. Isolation and Purification

After 7 days, the cultured rice with the fungus was cut into small fragments, which was then subsequently extracted with EtOAc by ultrasonication three times. The solvent was then removed under reduced pressure under vacuum to yield the total extract (4.7 g). The crude extract was fractionated by ODS flash column chromatography (5 × 30 cm), eluting with 2L each of MeOH-H_2_O (20:80, 40:60, 60:40, 80:20, 100:0). The fraction eluted with 80% MeOH was subjected to Sephadex LH-20 (CH_2_Cl_2_: MeOH 1:1) to obtain five subfractions (A–E), and subfraction B was subsequently purified by HPLC (Kromasil Eternity XT-5-C18 column, 250 × 10 mm i.d., 5 μm, 2mL min^−1^), with gradient elution from 75% to 85% MeOH in H_2_O with 0.2% AcOH to afford compounds **2** (t_R_ = 23.0 min, 2.5 mg) and **1** (t_R_ = 45.0 min, 1.6 mg).

The fraction of MeOH-H_2_O (60:40) was chromatographed on Sephadex LH-20 (CH_2_Cl_2_-MeOH, 1:1) to yield five subfractions (A–D). Fraction D was then isolated by silica gel column chromatography (CC) (200−300 mesh), eluting with a CH_2_Cl_2_-acetone gradient system (50:1, 20:1, 10:1, 6:1, 3:1, 2:1, 1:1) to yield five subfractions. The five fractions were analyzed by HPLC, revealing that compounds **3** and **4** were mainly detected in fraction 4.1. Fraction 4.1 was further purified by HPLC (ACN/H_2_O, 45%/55%), which created compounds **3** (2 mg) and **4** (3.5 mg). 

Compound **1**: white solid; ${\left[\alpha \right]}_{D}^{25}$ −20.5 (*c* 0.10, MeOH); CD (MeOH) 300 (*Δε* –0.97) nm, 270 (*Δε* +1.33) nm, 226 (*Δε* –2.78) nm, 211 (*Δε* +2.14) nm. ^1^H NMR (600 MHz, DMSO-*d6*) and ^13^C NMR (150 MHz, DMSO-*d6*) data (Table 1 and Table 2). ESIMS *m/z* 471 [M + H]^+^; HR-ESIMS *m/z* 528.2670 [M + H]^+^, calcd for C_28_H_39_O_6_, 528.2625.

Compound **2**: white solid; ${\left[\alpha \right]}_{D}^{25}$ 35.6 (*c* 0.10, MeOH); CD (MeOH) 224 (*Δε* –6.59) nm, 289 (*Δε* –4.48) nm. ^1^H NMR (600 MHz, DMSO-*d6*) and ^13^C NMR (150 MHz, DMSO-*d6*) data (Table 1 and Table 2). HR-ESIMS *m/z* 487.2964 [M + H]^+^, calcd for C_31_H_38_N_2_O_3_, 487.2966.

Compound **3**: white solid; ${\left[\alpha \right]}_{D}^{25}$ −77.1(*c* 0.10, MeOH); CD (MeOH) 225 (*Δε* –0.74) nm, 245 (*Δε* –0.54) nm, 290 (*Δε* –0.10) nm. ^1^H NMR (600 MHz, DMSO-*d6*) and ^13^C NMR (150 MHz, DMSO-*d6*) data (Table 1 and Table 2). HR-ESIMS *m/z* 589.3296 [M + H]^+^, calcd for C_34_H_40_N_2_O_7_, 589.3321.

Compound **4**: white solid; ${\left[\alpha \right]}_{D}^{25}$ −12.5 (*c* 0.10, MeOH); CD (MeOH) 219 (*Δε +*3.97) nm, 236 (*Δε* +3.46) nm, 303 (*Δε* –0.60) nm. ^1^H NMR (600 MHz, DMSO-*d6*) and ^13^C NMR (150 MHz, DMSO-*d6*) data (Table 1 and Table 2). HR-ESIMS *m/z* 513.2741 [M + H]^+^, calcd for C_32_H_36_N_2_O_4_, 513.2748.

### 3.5. Preparation of MTPA Esters of **2c** and **2d**

Compound **2** (200 μg for each) was reacted with either *R*-(−)- or *S*-(+)-MTPA Cl (5 μL) in 50 μL of CH_2_Cl_2_ and 50 μL of pyridine for 2 h. The reaction mixture was diluted with H_2_O and extracted with CH_2_Cl_2_ three times. The organic layers were combined and separated by HPLC (Cosmosil ARII, 10 × 50 mm; 20−80% MeCN in H_2_O) to afford the *S*-(−)- or *R*-(+)-MTPA esters, **2c** and **2d**.

^1^H NMR data of **2c** (600 MHz in DMSO-d6): δ_H_ 10.81 (s, NH-1’), 7.93 (s, NH-2), 7.45 (d, J = 7.8 Hz, H-4’), 7.27 (d, 8.0, H-7’), 7.06 (s, H-2’), 7.02 (m, H-6’), 6.97 (m, H-5’), 6.94 (d, J = 15.1 Hz, H-22), 6.32 (m, H-21), 6.18 (m, H-13), 5.25 (brs, H-7), 5.13 (m, H-14), 3.88 (m, H-19), 3.81 (m H-3), 2.84 (m, H-4), 2.71 (m, H-10a), 2.65 (m, H-8), 2.58 (m, H-10b), 2.27 (m, H-20a), 2.18 (m, H-5), 2.07 (m, H-20b), 2.02 (m, H-16), 1.98 (m, H-15a), 1.64 (m, H-15b), 1.36 (m, H-18a), 1.27 (s, Me-12), 1.20 (m, H-17a), 1.06 (m, H-18b), 0.89 (d, J = 6.7 Hz, Me-16), 0.83 (m, H-17b), 0.80 (d, J = 6.6 Hz, Me-11). HRESI MS: m/z 703.1705 [M + H]^+^.

^1^H NMR data of **2d** (600 MHz in DMSO-d6): δ_H_ 10.91 (s, NH-1’), 8.20 (s, NH-2), 7.46(d, J = 7.6 Hz, H-4’), 7.27 (d, J = 7.9 Hz,0, H-7’), 7.06 (s, H-2’), 7.03 (m, H-6’), 6.97 (m, H-5’), 6.95 (d, J = 14.9 Hz, H-22), 6.36 (m, H-21), 6.13 (m, H-13), 5.24 (brs, H-7), 5.12 (m, H-14), 4.07 (m, H-19), 3.83 (m H-3), 2.85 (m, H-4), 2.71 (m, H-10a), 2.67 (m, H-8), 2.60 (m, H-10b), 2.49 (m, H-20a), 2.18 (m, H-5), 2.16 (m, H-20b), 2.01 (m, H-16), 1.98 (m, H-15a), 1.64 (m, H-15b), 1.27 (s, Me-12), 1.21 (m, H-18a), 1.18 (m, H-17a), 1.03 (m, H-18b), 0.89 (d, J = 6.7 Hz, Me-16), 0.82 (m, H-17b), 0.80 (d, J = 6.6 Hz, Me-11). HRESI MS: m/z 703.1738 [M + H]^+^.

### 3.6. Computational of ECD

A conformational search was carried out in the MMFF94 molecular mechanics force field using the MOE software package [22], and all the conformers within an energy window of 10 kcal/mol were regarded as the initial conformations. The geometry optimization and frequency calculations were performed with Gaussian16 RevB.01 [23], using the ωB97XD or B3LYP functional at the 6-311G(d,p) level of theory to verify the stability and obtain the energies at 298.15 K and a 1 atm pressure. The Boltzmann distribution was calculated according to their Gibbs free energies. ECD calculations were conducted by using the Cam-B3LYP functional at the TZVP level of theory. The Solvation Model Based on Density (SMD) was used as the solvation model. The Boltzmann-averaged ECD spectra were obtained by using SpecDis 1.71 software [24].

### 3.7. Cytotoxicity Assay

The cytotoxic activities of **1**–**16** against human carcinoma cells HeLa and A549 were determined using the 3-(4,5-dimethyl-2-thiazolyl)-2,5-diphenyl-2-H-tetrazolium bromide (MTT) assay. The cells were maintained in RPMI-1640 containing 10% (*v/v*) fetal bovine serum (FBS) and 0.4% (*v/v*) penicillin–streptomycin solution (10,000 units/mL penicillin and 10,000 μg/mL streptomycin, 100×) at 37 °C under 5% CO_2_. The cells were digested by trypsinization and then diluted to a concentration of 1 × 10^4^ cells/mL. The diluted cell suspensions were then placed into 96-well microtiter plates and incubated with the test samples for 72 h. The control contained 2 μL MeOH. After incubation, the MTT solution was added and plates were incubated for 4 h. The supernatant liquid was removed, and the cells were disrupted with 150 μL DMSO for 10 min. The absorption was measured at 570 nm.

## 4. Conclusions

Four new cytochalasans, phychaetoglobins A–D (**1**–**4**), together with twelve known cytochalasans (**5**–**16**), were identified from the culture of *C. globosum* kz-19. All cytochalasans were tested for their cytotoxicities against HeLa and A549 tumor cell lines. Among compounds **4**–**8**, those that contained a C-5/C-6 double bond and a C-7 hydroxyl group or a C-6/C-7 epoxide ring exhibited strong cytotoxic activities with the IC_50_ values of 3.7–13.7 μM. Compound **1** represents the first example of a cytochalasan with a five-membered lactone fused in the macrocyclic ring. Natural cytochalasans containing a cyclopentenone or a pyrrole moiety in the macrocyclic ring are rare. To our knowledge, only four natural cytochalasans containing a cyclopentenone ring have been reported, including cytoglobosin A [4], chaetoglobosin U [25], V [17], and Vb [26]. Furthermore, 11 pyrrole-containing cytochalasans are penochalasins A1-C3 [27] and armochaetoglobosins K-R [28]. Therefore, this study diversifies the structures of natural cytochalasans.

## Figures and Tables

**Figure 1 marinedrugs-19-00438-f001:**
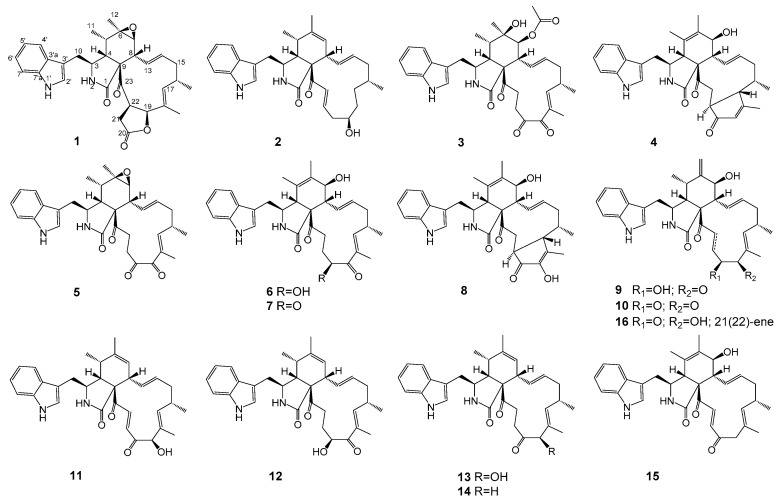
Chemical structures of compounds **1**–**16**.

**Figure 2 marinedrugs-19-00438-f002:**
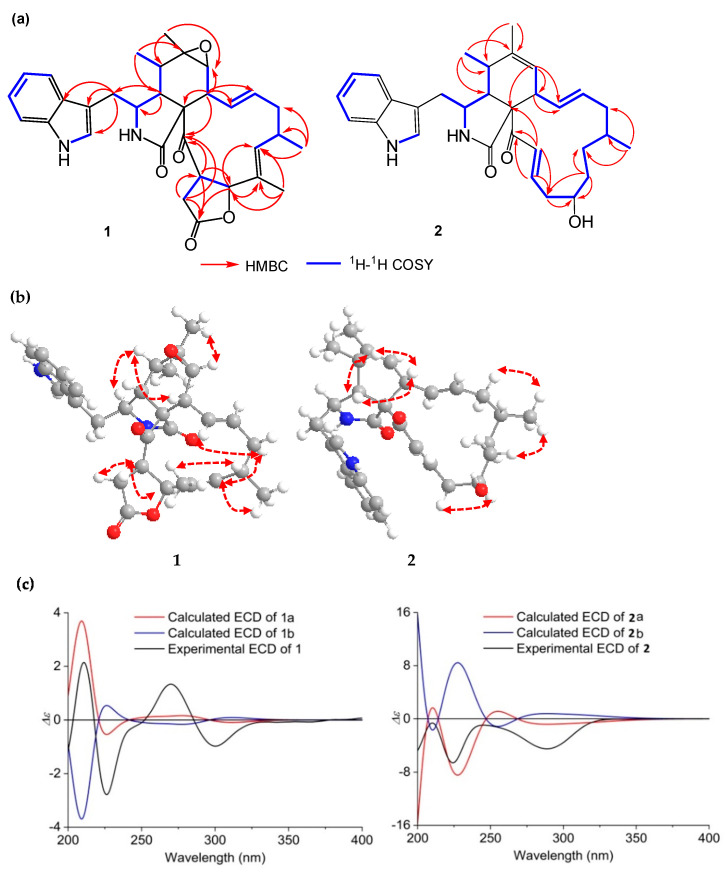
(**a**) Key HMBC and ^1^H-^1^H COSY correlations of **1** and **2**; (**b**) Key NOESY correlations of **1** and **2**; (**c**) Experimental and calculated ECD of **1** and **2** (**1a**, **1b** and **2a**, **2b**) in MeOH.

**Figure 3 marinedrugs-19-00438-f003:**
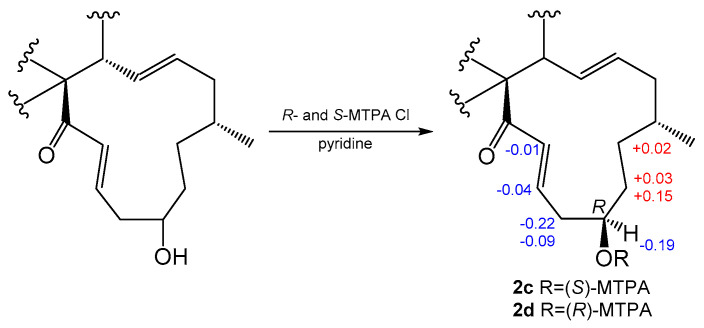
Values of Δ*δ_S_*-*δ_R_* of the MTPA esters of **2** (in DMSO-*d6*).

**Table 1 marinedrugs-19-00438-t001:** ^1^H NMR Data for Compounds **1**–**4** recorded in DMSO-*d6* (600 MHz, *J* in Hz).

Position	1	2	3	4
2-NH	8.52, s	7.86, s	8.50, s	8.21, s
3	3.84, dd (8.6, 4.3)	3.66, dd (8.5, 4.2)	3.72, dd (8.5, 4.2)	3.36, m
4	1.90, m	2.83, m	2.14, m	2.95, brs
5	1.47, m	2.20, m	1.90, m	
7	2.83, d (5.2)	5.24, s	4.47, d (12.2)	3.85, d (9.7)
8	2.61, dd (10.4, 5.2)	2.61, m	2.91, dd (12.2, 10.2)	2.12, t (10.4)
10	2.87, dd (15.0, 3.6); 2.95, dd (15.0, 5.0)	2.57, dd (14.3, 6.6); 2.68, dd (14.3, 4.9)	2.93, m 2.69, dd (15.0,3.8)	2.73, dd (14.0, 5.4) 2.28, m
11	0.99, d (7.4)	0.76, d ( 7.3)	0.98, d (7.0)	1.08, s
12	1.29, s	1.62, s	1.03, s	1.51, s
13	5.43, dd (15.2, 10.4)	6.15, dd (15.0, 9.5)	5.53, dd (14.9, 9.5)	5.17, dd (14.9, 10.0)
14	4.78, ddd (15.2, 11.2, 3.6)	5.12, ddd (15.0, 10.6, 2.9)	4.86 ddd (15.0, 11.2, 2.8)	6.25, dd (14.9, 10.6)
15	1.54, dd (11.8, 11.6) 2.19, m	1.62, m 2.00, m	2.25, m 1.67, m	2.53, m 1.76, dd (13.3, 11.5)
16	2.32, m	2.19, m	2.64, m	2.37, m
16-CH_3_	0.88, d (6.5)	0.86, d (6.7)	0.92, d (6.7)	0.67, d (6.8)
17	5.13, d (11.1)	1.28, m 0.80, m	5.85, dd (10.1, 1.3)	2.95, brs
18	--	1.38, m 1.10, m	--	--
18-CH_3_	1.23, s	--	1.70, s	2.09, s
19	4.84, d (8.8)	3.65, m	--	5.59, s
20	--	2.29, m 2.20, m	--	--
21	2.46, m	6.69, ddd (15.9, 9.8, 5.5)	2.35, m 1.74, m	2.43, m
22	2.66, m	6.85, d (15.9)	2.45, m	3.54, m 2.45, m
25	--	--	1.90, s	--
1’-NH	11.1, s	10.8, s	11.0, s	10.9, s
2’	7.26, d (2.2)	7.04, s	7.17, d (1.8)	7.02, s
4’	7.56, d (7.7)	7.45, d (8.0)	7.54, d (7.8)	7.43, d (7.9)
5’	7.02, m	6.95, m	7.00, m	6.96, m
6’	7.14, m	7.03, m	7.07, m	7.06, m
7’	7.29, d (7.9)	7.29, d (8.1)	7.34, d (7.8)	7.33, d (8.1)

**Table 2 marinedrugs-19-00438-t002:** ^13^C NMR Data for Compounds **1**–**4** recorded in DMSO-*d6* (150 MHz).

Position	1	2	3	4
1	172.8	174.0	173.4	174.0
3	53.0	53.3	52.3	57.1
4	48.2	48.5	43.7	48.7
5	35.3	34.5	39.2	125.7
6	56.7	139.8	71.9	133.8
7	59.9	126.3	73.0	67.8
8	44.7	47.3	43.3	53.6
9	65.8	66.4	62.1	63.0
10	31.9	33.5	31.2	31.8
11	13.9	13.3	12.9	17.0
12	20.1	19.8	24.3	14.5
13	130.1	130.3	125.8	131.4
14	130.9	132.4	133.3	132.1
15	40.4	41.4	40.1	38.1
16	31.5	33.8	32.2	31.0
16-CH_3_	20.2	22.4	19.2	15.5
17	137.8	29.6	154.8	47.6
18	129.0	34.8	131.0	180.5
18-CH_3_	11.4	--	10.4	17.2
19	84.9	69.2	195.8	128.5
20	175.0	39.4	204.9	207.0
21	31.5	142.8	32.7	42.6
22	49.3	129.3	35.9	40.5
23	204.7	197.3	207.3	209.7
24	--	--	170.0	--
25	--	--	20.7	--
2’	125.8	124.2	125.5	123.6
3’	108.0	109.8	107.9	110.0
3a’	127.8	127.9	127.9	127.1
4’	118.2	118.5	118.4	118.1
5’	118.9	118.7	118.8	118.5
6’	121.1	121.2	121.0	121.0
7’	111.7	111.7	111.4	111.5
7a’	136.1	136.4	135.9	136.2

**Table 3 marinedrugs-19-00438-t003:** Cytotoxicities of Compounds **1**–**16** against HeLa and A549 cell lines ^a^.

Cells	IC_50_ (μM) Values of Compounds								
	1	2	3	4	5	6	7	8	Adriamycin
HeLa	--^b^	23.9 ± 0.4	16.1 ± 0.3	9.2 ± 0.3	10.5 ± 0.1	7.5 ± 0.2	3.7 ± 0.3	3.8 ± 0.3	0.8 ± 0.3
A549	32.3 ± 0.2	-- ^b^	22.3 ± 0.4	13.7 ± 0.2	7.6 ± 0.2	12.3 ± 0.3	7.3 ± 0.5	11.0 ± 0.2	2.9 ± 0.2
	**9**	**10**	**11**	**12**	**13**	**14**	**15**	**16**	
HeLa	39.1 ± 0.5	26.0 ± 0.4	21.6 ± 0.2	32.2 ± 0.2	29.3 ± 0.2	25.6 ± 0.2	12.2 ± 0.1	33.7 ± 0.3	
A549	-- ^b^	23.2 ± 0.3	13.4 ± 0.1	-- ^b^	22.5 ± 0.3	14.9 ± 0.2	17.3 ± 0.3	25.9 ± 0.2	

^a^ Adriamycin and DMSO were used as positive and negative controls, and the data were expressed as the means ± SD (*n* = 3); ^b^ IC_50_ values were more than 40 μM.

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
