# Peer review of "New Cytotoxic Cytochalasans from a Plant-Associated Fungus Chaetomium globosum kz-19"

_marinedrugs, 2021, doi:10.3390/md19080438_

Round 1

Reviewer 1 Report

The authors reported new cytotoxic cytochalasans from a mangrove-associated fungus Chaetomium globosum kz-19. The contents are well-organized and sound. But there are some issues to be revised before the publication.

1) line 47, prochaetoglobosin IIIed?? (15)

2) lines 79-80, authors indicated that the NOESY correlations from H-22 to H-23 and from H-23 to H-21α revealed 79 the β-orientation of the lactone ring. However, C-23 has no proton (C-23 is ketone position). Please check it again and correct.

3) Authors should suggest how to assign the stereochemistry of C-16 in compound 1; there is no information for the assignment of methyl group at C-16.

4) Authors should indicate exactly the structures of 1a and 1b for ECD calculations    

5) Why don’t you change the position of ECD calculations of 1 and 2. The left position should be the data of 1.

6) In compound 2, I don’t think that 2a and 2b for Mosher’s reaction are the same structures to those that you used for ECD calculations. Author should suggest structures of two conformers that you used for ECD calculations, and also change the codes to others to be distinguishable.

7) The Figure 3 caption should be corrected; to “values of ΔdS-dR of”

8) Authors should indicate all other structures of conformers that you used for ECD calculations for compounds 3 and 4.

9) in Table 1, Authors should check the NMR data again to see if there are mistakes. There are some issues where coupling constants are not correct; in title, Compounds 14 -> Compounds 1-4

10) There are many typos and space errors. Please check them and correct.

Author Response

Manuscript Number: marinedrugs-1317021

TITLE: New Cytotoxic Cytochalasans from A Plant-Associated Fungus Chaetomium globosum kz-19

Dear editor:

We thank the reviewers for their insightful comments. We have rewritten and addressed the reviewer’s issues thoroughly with blue words in the revised edition. We believe that the changes made according to their suggestions have significantly strengthened the manuscript.

Reviewer #1:

The authors reported new cytotoxic cytochalasans from a mangrove-associated fungus Chaetomium globosum kz-19. The contents are well-organized and sound. But there are some issues to be revised before the publication.

1) line 47, prochaetoglobosin IIIed?? (15)

Response: We have corrected the name of 15 by using subscript as prochaetoglobosin IIIed.

2) lines 79-80, authors indicated that the NOESY correlations from H-22 to H-23 and from H-23 to H-21α revealed 79 the β-orientation of the lactone ring. However, C-23 has no proton (C-23 is ketone position). Please check it again and correct.

Response: Thanks for your suggestion. We have deleted the wrong correlations and revise sentence as follows. ‘Additionally, the NOESY correlations of H-22 with H-19 and H-21α, together with the coupling constant (J = 8.8 Hz) between H-19 and H-22 revealed the β-orientation of the five-numbered lactone ring.’

3) Authors should suggest how to assign the stereochemistry of C-16 in compound 1; there is no information for the assignment of methyl group at C-16.

Response: Thank you for your suggestion. We have added the elucidation of the configuration at C-16. Please check the following sentence. ‘The observed NOESY cross-peaks of H-15α with H-13 and H-17, CH3-18 with H-16, as well as H-17 with CH3-16 implied an α orientation at CH3-16 and the E-geometry of Δ17-double bond.’

4) Authors should indicate exactly the structures of 1a and 1b for ECD calculations.

Response: Thank you for your suggestion. 1b is the enantiomer of 1a. We have indicated it in the manuscript and supplemented the figures of the optimized conformation of 1 in supporting information.

5) Why don’t you change the position of ECD calculations of 1 and 2. The left position should be the data of 1.

Response: Thank you for your suggestion. We have revised the ECD figure.

6) In compound 2, I don’t think that 2a and 2b for Mosher’s reaction are the same structures to those that you used for ECD calculations. Author should suggest structures of two conformers that you used for ECD calculations, and also change the codes to others to be distinguishable.

Response: Thank you for your suggestion. We’ve kept the name of the 2 products of Mosher’s reaction, and changed the name of the two conformers as 2c and 2d, respectively.

7) The Figure 3 caption should be corrected; to “values of ΔdS-dR of”

Response: Done.

8) Authors should indicate all other structures of conformers that you used for ECD calculations for compounds 3 and 4.

Response: Thank you for your suggestion. We have added all the conformers of compounds 1-4, and showed the details of each conformer in the supporting information. In addition, we added the contents of ECD determination in the experiment section.

9) in Table 1, Authors should check the NMR data again to see if there are mistakes. There are some issues where coupling constants are not correct; in title, Compounds 14 -> Compounds 1-4

Response: Thanks for your suggestions. We have corrected some of the 1H NMR and 13C NMR data in the Tables 1 and 2, and marked in blue.

10) There are many typos and space errors. Please check them and correct.

Response: We have checked and revised some type errors in the manuscript.

Reviewer 2 Report

The authors obtained a fungal strain from the mangrove and isolated sixteen cytochalasans including four new compounds. The structures of newly isolated compounds were elucidated by spectroscopic data analyses in combination with ECD calculation and chemical derivatization method. However, there are several issues were found, and they need to be clarified first before further consideration.

  1. Recently IC50 values over 10 uM is not accepted as real cytotoxicity. Therefore, compound 15 could be eliminated from the highly active compounds in the abstract and the manuscript. Also, “significant cytotoxicity” will need the statistic support of significance. If the description is supported by statistic analysis, please provide the evidence. Otherwise, the authors need to find the proper term to describe the potency of the activity.
  2. H-4’/H-5’/H-6’/H-7’ might be in a spin system, and the adjacent protons will be coupled each other. However, the measured coupling constants were found to be inconsistent. It would be highly recommended to measure precise coupling constants with NMR data processing.
  3. In line 58: the authors used the term “secondary methyl”. However, the term “secondary” and “methyl” seems to be incorrectly used. “Secondary” is used for the functional group attached to two of alkyl branches, but “methyl group” can have only one connection. Please use the appropriate term.
  4. In lines 55-64: The authors mentioned the structural similarity between compound 1 and 5, based on the 1D NMR data. However, the number of carbonyls and oxygenated sp3 carbons as well as the presence of lactone are different. If the authors plan to describe the structure of 1 from the similarities in 1D NMR data of 1 with those of 5, please indicate the similar features selectively.
  5. In Figure 2a: the legend for the 2D NMR correlations is missing. The authors need to address the COSY (or TOCSY or any correlations) and HMBC correlations.
  6. In Figure 2c: The ECD spectra related to compound 2 were depicted prior to those of compound 1. It doesn’t match with the flow of description.
  7. In ECD calculation, the authors did not provide the detail parameters of calculations including softwares. Furthermore, the authors find only one conformer of the compound and its enantiomeric pair. It is absolutely not true. If the authors considered the conformers more than one, please provide details. Otherwise, the authors need to perform conformer study.
  8. In the main text, the authors described the cytotoxicity of the isolates were evaluated against two cell lines. However, in the experimental sections, the authors mentioned three strains. Please add the data even though they are negative.
  9. Figure 3: “R/S” should be italicized. Also, a configuration information on the Mosher’s esters seems to be unexpectedly missing. Please modify the figure.
  10. In Tables 1 and 2: chemical shift values are not matching with supplementary spectroscopic data. Please inspect chemical shift and coupling constants carefully and revise the values in the table and manuscript.
  11. In Figures S8, T1 noise was found intense. Please follow data process to eliminate the noise signal to show key correlations distinctively.

Author Response

Manuscript Number: marinedrugs-1317021

TITLE: New Cytotoxic Cytochalasans from A Plant-Associated Fungus Chaetomium globosum kz-19

Dear editor:

We thank the reviewers for their insightful comments. We have rewritten and addressed the reviewer’s issues thoroughly with blue words in the revised edition. We believe that the changes made according to their suggestions have significantly strengthened the manuscript.

Reviewer 2.

The authors obtained a fungal strain from the mangrove and isolated sixteen cytochalasans including four new compounds. The structures of newly isolated compounds were elucidated by spectroscopic data analyses in combination with ECD calculation and chemical derivatization method. However, there are several issues were found, and they need to be clarified first before further consideration.

  1. Recently IC50 values over 10 uM is not accepted as real cytotoxicity. Therefore, compound 15 could be eliminated from the highly active compounds in the abstract and the manuscript. Also, “significant cytotoxicity” will need the statistic support of significance. If the description is supported by statistic analysis, please provide the evidence. Otherwise, the authors need to find the proper term to describe the potency of the activity.

Response: Thanks for your suggestions. We have revised the description as ‘moderate cytotoxicity’ for the active compounds. Additionally, compound 15 has been deleted from the highly active compounds in the abstract and the manuscript. The description of the statistic analysis for the assay has been supplemented below Table 3.

  1. H-4’/H-5’/H-6’/H-7’ might be in a spin system, and the adjacent protons will be coupled each other. However, the measured coupling constants were found to be inconsistent. It would be highly recommended to measure precise coupling constants with NMR data processing.

Response: We have checked the coupling constants from H-4’ to H-7’, and revised the incorrect constants in Table 1.

  1. In line 58: the authors used the term “secondary methyl”. However, the term “secondary” and “methyl” seems to be incorrectly used. “Secondary” is used for the functional group attached to two of alkyl branches, but “methyl group” can have only one connection. Please use the appropriate term.

Response: We have revised the description for the signals of the methyl group in the manuscript as follows.

 ‘two methyl singlets (δH 1.23 and 1.29) and two methyl doublets at δH 0.88 (d, 6.5 Hz) and 0.99 (d, 7.4 Hz).’

  1. In lines 55-64: The authors mentioned the structural similarity between compound 1 and 5, based on the 1D NMR data. However, the number of carbonyls and oxygenated sp3 carbons as well as the presence of lactone are different. If the authors plan to describe the structure of 1 from the similarities in 1D NMR data of 1 with those of 5, please indicate the similar features selectively.

Response: We have revised the sentences for the elucidation of compound 1 and 5 as follows.

‘Further analyses of 1H and 13C NMR, as well as 2D NMR spectra of 1 (Figure 2A) supported the presence of an indolyl moiety, a pyrrolidine-2-one, and a cyclohexane ring in the structure, which were same as that of 5. However, the differences between 1 and 5 were that 1 possessed two less ketone groups and one more ester group than 5.’

  1. In Figure 2a: the legend for the 2D NMR correlations is missing. The authors need to address the COSY (or TOCSY or any correlations) and HMBC correlations.

Response: We have added the address for the COSY and HMBC correlations in Figure 2a.

  1. In Figure 2c: The ECD spectra related to compound 2 were depicted prior to those of compound 1. It doesn’t match with the flow of description.

Response: We have exchanged them.

  1. In ECD calculation, the authors did not provide the detail parameters of calculations including softwares. Furthermore, the authors find only one conformer of the compound and its enantiomeric pair. It is absolutely not true. If the authors considered the conformers more than one, please provide details. Otherwise, the authors need to perform conformer study.

Response: Thanks for your suggestions. We have supplemented the ECD calculation parameters and more details in the sections of ‘Results and discussion’ and ‘Materials and Methods’. For example in the results and discussion, ‘The Merck Molecular Force Field (MMFF) conformational search in an energy window of 10 kcal/mol, which were optimized at the ωB97X/TXVP level (in MeOH) yielding 2 low energy geometries over 1% Boltzmann population (Figure S9).’

We also added more details of ECD conformer and calculation for each new compound in the supporting information. Please see the SI materials, the revised contents are such as ‘TDDFT theory, ωB3LYP functional and 6-311G(d,p) level of theory, methanol as solvent for structural optimization, compound 1 have 9 conformations, of which 2 conformations have Boltzmann content >1%, listed in the table. The calculated result of 1 is consistent with the experimental result, and the absolute configuration of 1 is confirmed as shown in the figure below (s=0.30eV).’

  1. In the main text, the authors described the cytotoxicity of the isolates were evaluated against two cell lines. However, in the experimental sections, the authors mentioned three strains. Please add the data even though they are negative.

Response: We have revised the number of the tested cell lines, because we used only 2 cell lines for testing the cytotoxicity of all cytochalasans.

  1. Figure 3: “R/S” should be italicized. Also, a configuration information on the Mosher’s esters seems to be unexpectedly missing. Please modify the figure.

Response: We have revised according to your suggestion. Thanks!

  1. In Tables 1 and 2: chemical shift values are not matching with supplementary spectroscopic data. Please inspect chemical shift and coupling constants carefully and revise the values in the table and manuscript.

Response: We have re-attached the spectroscopic figures of all new compounds, and corrected some differences between the data in Tables and those in SI.

  1. In Figures S8, T1 noise was found intense. Please follow data process to eliminate the noise signal to show key correlations distinctively.

Response: We have re-processed the data.

Reviewer 3 Report

General remarks:

This is a paper dealing with the isolation and identification of sixteen compounds, four of them new in the literature. Structures are assigned and characterized, even though IR data and melting points are missing for the new compounds (which can be accounted for by the small amount isolated in the case of melting points that require a previous recrystalization). The Introduction, somewhat short, could be improved. Discussion for compounds 1 and 2 needs to be improved as does the discussion for the Biological assays.

The overall quality of the work is not very good: several mistaken values and exchanged a/b orientations are written in the text, and the presentation of the supplementary material must be improved...there is no point in showing spectra you cannot read.

Author Response

Manuscript Number: marinedrugs-1317021

TITLE: New Cytotoxic Cytochalasans from A Plant-Associated Fungus Chaetomium globosum kz-19

Dear editor:

We thank the reviewers for their insightful comments. We have rewritten and addressed the reviewer’s issues thoroughly with blue words in the revised edition. We believe that the changes made according to their suggestions have significantly strengthened the manuscript.

Reviewer 3.

This is a paper dealing with the isolation and identification of sixteen compounds, four of them new in the literature. Structures are assigned and characterized, even though IR data and melting points are missing for the new compounds (which can be accounted for by the small amount isolated in the case of melting points that require a previous recrystalization). The Introduction, somewhat short, could be improved. Discussion for compounds 1 and 2 needs to be improved as does the discussion for the Biological assays. The overall quality of the work is not very good: several mistaken values and exchanged a/a orientations are written in the text, and the presentation of the supplementary material must be improved...there is no point in showing spectra you cannot read.

Response: Thanks for your suggestions. As you suggested, we couldn’t measure IR and melting point for each new compound because of the small amount of the samples. In addition, we have added antitumor pharmacological contents in the introduction section as follows (the first sentence). We also revised the discussion for the biological assays according to your suggestions. Please see the second and the third sentences.

  1. ‘In terms of anti-tumor pharmacological effects, many researchers found that cytochalasans could regulate the morphology and phenotype of tumor cells and induce apoptosis [12]. Thus, cytochalasins are potential to be used as a supplementary chemotherapeutic drug and decrease the incidence of drug resistance clinically in the future [13].’
  2. ‘Compounds 5 and 7 exhibited the strongest cytotoxicity against A549 cell lines with the IC50 values below 10 m Furthermore, compounds 4-8 showed the moderate cytotoxicity against HeLa cell line, with IC50 values of 3.7~10.5 mM, implying that the moieties of a double bond at C-5/C-6 and a hydroxyl group at C-7 or an epoxide ring at C-6/C-7 were the important characteristics responsible for the cytotoxicity.’
  3. Coumpound 2 showed weak cytotoxicities, as it did not possess the functional groups related to the bioactivity.

English suggestions:

Instead of ‘---‘ – use ‘---'

Line 42 – replace oxygated with oxygenated

Response: Done.

Line 107 – ‘the NOESY correlations’ – ‘NOESY correlations’

Response: Done.

Line123 – ‘differences compared to 7 were’ – ‘when compared to 7, differences were’

Response: Done.

Line 173 – Correct Angilent to Agilent

Response: Done.

  1. Introduction

Line 31 – I believe you mean 11 or 13 member macrocycles

Response: We have revised the sentence as ‘The polyketo-amino acid hybrid structure of cytochalasans forms different fused macrocycles (typical 11 or 13-membered macrocycles) of cytochalasans, …’.

Line 47 – Correct prochaetoglobusin IIIed to prochaetoglobusin III

Response: The name of this compound should be ‘prochaetoglobusin IIIed.’ We have revised it.

  1. Results

2.1 Structure elucidation

Lines 55-59 – these lines need to be reviewed – the values are not in accordance with table 1.

Response: Done.

Insert multiplicity and J value for the secondary methyls at 0.88 and 0.99.

Response: Done.

Line 61 – Correct 175.0 to 174.9 (table 2) 2

Response: Done.

Line 65 – there’s no pyrrole ring in the structure

Response: we have deleted ‘pyrrole ring’.

Line 71 – correct ‘two trans olefinic bonds’ to ‘two olefinic bonds’. You cannot tell they are trans by HMBC.

Response: We have deleted ‘trans’ in the sentence.

Line 79 -This sentence makes no sense – there’s no H-23.

Response: We have revised the sentence as follows and deleted the wrong correlation of H-23.

‘Additionally, the NOESY correlations of H-22 with H-19 and H-21α, together with the coupling constant (J = 8.8 Hz) between H-19 and H-22 revealed the β-orientation of the five-numbered lactone ring.’

Figure 2 – exchange the order of the graphics (data for structure 1 should come first)

Response: Done.

LIne 103 – what’s the geometry of this double bond?

Response: We have added the elucidation of the geometry of the double bonds at C-17/18 in the manuscript. Please see the following sentence.

‘The observed NOESY cross-peaks of H-15α with H-13 and H-17, CH3-18 with H-16, as well as H-17 with CH3-16 implied an α orientation at CH3-16 and the E-geometry of Δ17-double bond.’

Line 105 – Me-11 is not a in the drawing, nor in structure 1, Refer to Lines 77/78

Response: We have revised the sentence of ‘NOESY correlations of Me-11 with H-3 and Me-12 with H-7 ...’, and changed another drawing in Figure 2b.

Line 128 – H-4, H-5 and H-8 are a in the drawing

Response: H-4, H-5, and H-8 should be α orientation. We have change the figures of NOESY correlations in figure 2b.

Tables 1 and 2 – insert solvent used in the legend

Response: Done.

2.2 Biological assay

Line 149 - Correct Hela to HeLa

Response: Done.

Table 3 – Legend and table, correct Hela to HeLa

Response: Done.

Line 152 – Values are 3.7-12.2 for HeLa cells and 7.3-17.3 for A549. These results should be discussed separately.

Response: Done.

Compounds 5 and 7 exhibited the strongest cytotoxicity against A549 cell lines with the IC50 values below 10 mM. Furthermore, compounds 4-8 showed the moderate cytotoxicity against HeLa cell line, with IC50 values of 3.7~10.5 mM, implying that the moieties of a double bond at C-5/C-6 and a hydroxyl group at C-7 or an epoxide ring at C-6/C-7 were the important characteristics responsible for the cytotoxicity.

Lines 154 to 158 – I don’t agree with these sentences and cannot understand what you mean.

Response: We have corrected this sentence.

‘Compounds 3, 10-11, and 13-16 exhibited cytotoxic activity (IC50 values of 12.2~33.7 mM), indicating that the presence of a ketone group other than a hydroxyl group at C-20 could increase the bioactivity.’

  1. Materials and methods:

No corrections

  1. Conclusions:

Line 244 – Correct Hela to HeLa.

Response: Done.

Conflicts of interest

Correct the sentence

Response: Done.

  1. References

Line 44, chaetoglobosin E – replace ref 14 with ref 13.

Line 46, chaetoglobosin J - insert ref 21

Line 48, chaetoglobosin D - insert ref 13

Reorder the references, so that successive numbers appear in the text.

Response: We have revised the references and re-order the numbers of references.

Supplementary material

Replace spectra S8, S17, S28, S29, S36, S37 and S38 with medium quality spectra where you can read the information (especially mass spectra).

Response: Done.

Replace TCOSY with TOCSY, both in the index and in the legends of the spectra.

Response: Done.

Fig S21 – Correct legend to 1H NMR spectrum of 2b. After correction, and also for Fig S29, insert solvent and frequence of the spectrum in the legend.

Response: Done.

Fig S37 – Correct to NOESY spectrum in the index

Correct page number for Fig S9 – correct number is page 6.

Correct all the following page numbers in the index.

Response: Done.

Reviewer 4 Report

This manuscript is not recommended for publication. Because there are many deficiencies in the content of this manuscript. Below is a list of comments on this manuscript.

  1. Page 2, lines 68-71; For Compound 1, the authors should explain why the HMBC spectrum reveals the presence of two trans olefinic bonds. From this data, it is impossible to make geometric isomer a trans.
  2. Page 2, line 75-76; The authors have determined from the NOE between H-8 and H-5 that the cyclohexane ring of compound 1 is a twist-boat conformation. However, that alone lacks grounds. It should be discussed in detail, including the analysis of the 1H-NMR J values of compound 1.
  3. Page 3, line 79; Since C-23 of compound 1 is a carbonyl group and does not have hydrogen, the notation of H-23 is incorrect. The authors should confirm this point.
  4. Page 3, line 82; The authors describe the conformations of 1a and 1b, but these conformations cannot be understood because their figures do not exist. The authors should add a figure of each conformation.
  5. Page 3, line 80; The determination of the stereochemistry of the lactone ring of compound 1 should be considered using the J values of the 1H-NMR spectrum.
  6. Page 3, line87; The conformations of compounds 1 and 2 are difficult to understand and should be modified.
  7. Page 4, line 111; Are they the same for MTPA esters (2a and 2b) and 2a and 2b described in the ECD spectrum of figure 2? Compound numbers are confused. The authors should correct this point.
  8. Page 4, line 118; Is DMSO-d6 suitable as a 1H-NMR measurement solvent for MTPA esters? This method relies on the conformation of the MTPA ester in solution. In addition, the original paper recommends the use of CDCl3. The authors should comment on this point.
  9. The authors should add information on the solvents used in the NMR measurements to Tables 1 and 2.
  10. Page 7, line 163; I can't find the information about the equipment used to measure the ECD spectrum. In addition, although the information on the measuring equipment of the UV spectrum is described, there is no description about the data in the text.
  11. Page 8, line 217; There is no experimental description of the conversion of compound 2 to MTPA ester. Also, there is no information on the spectral data of MTPA esters.

Author Response

Manuscript Number: marinedrugs-1317021

TITLE: New Cytotoxic Cytochalasans from A Plant-Associated Fungus Chaetomium globosum kz-19

Dear editor:

We thank the reviewers for their insightful comments. We have rewritten and addressed the reviewer’s issues thoroughly with blue words in the revised edition. We believe that the changes made according to their suggestions have significantly strengthened the manuscript.

Reviewer 4.

This manuscript is not recommended for publication. Because there are many deficiencies in the content of this manuscript. Below is a list of comments on this manuscript.

  1. Page 2, lines 68-71; For Compound 1, the authors should explain why the HMBC spectrum reveals the presence of two trans olefinic bonds. From this data, it is impossible to make geometric isomer a trans.

Response: Thanks for your suggestion. We have deleted the word ‘trans’ in the sentence.

  1. Page 2, line 75-76; The authors have determined from the NOE between H-8 and H-5 that the cyclohexane ring of compound 1 is a twist-boat conformation. However, that alone lacks grounds. It should be discussed in detail, including the analysis of the 1H-NMR J values of compound 1.

Response: Thanks for your suggestion. We have revised the correlations and added the coupling constants of H-8/H-13 and H-8/H-7. Please read the revised sentence as follows. ‘Additional coupling constants (J = 10.4, 5.2 Hz) of H-8 with H-13 and H-7 in the 1H NMR spectrum revealed that H-8 was orientated axial, and the NOESY cross-peaks of H-8 with H-4 and H-5 further confirmed the twist-boat conformation of the cyclohexane ring.’

  1. Page 3, line 79; Since C-23 of compound 1 is a carbonyl group and does not have hydrogen, the notation of H-23 is incorrect. The authors should confirm this point.

Response: We have revised the wrong sentence and deleted H-23. Please read the following sentence: ‘Additionally, the NOESY correlations of H-22 with H-19 and H-21α, together with the coupling constant (J = 8.8 Hz) between H-19 and H-22 revealed the β-orientation of the five-numbered lactone ring.’

  1. Page 3, line 82; The authors describe the conformations of 1aand 1b, but these conformations cannot be understood because their figures do not exist. The authors should add a figure of each conformation.

Response: 1b is the enantiomer of 1a. We have supplemented the conformations of each new compound for the ECD analysis in the supporting information.

  1. Page 3, line 80; The determination of the stereochemistry of the lactone ring of compound 1should be considered using the J values of the 1H-NMR spectrum.

Response: We have revised the elucidation for the stereochemistry of the lactone ring. Please see the above explanation of question 3.

  1. Page 3, line87; The conformations of compounds and 2are difficult to understand and should be modified.

Response: We have modified figure 2b to make clear the conformation of 1 and 2.

  1. Page 4, line 111; Are they the same for MTPA esters (2aand 2b) and 2a and 2b described in the ECD spectrum of figure 2? Compound numbers are confused. The authors should correct this point.

Response: We have re-named the products for MTPA esters as 2c and 2d and marked them in the manuscript.

  1. Page 4, line 118; Is DMSO-d6suitable as a 1H-NMR measurement solvent for MTPA esters? This method relies on the conformation of the MTPA ester in solution. In addition, the original paper recommends the use of CDCl3. The authors should comment on this point.

Response: Both MTPA esters of compound 2 were poor solubility in CDCl3 after the Mosher’s reaction. So, we used DMSO-d6 as the NMR solvent. We also can see the difference between the two products in the 1H NMR spectrum.

  1. The authors should add information on the solvents used in the NMR measurements to Tables 1 and 2.

Response: Done.

  1. Page 7, line 163; I can't find the information about the equipment used to measure the ECD spectrum. In addition, although the information on the measuring equipment of the UV spectrum is described, there is no description about the data in the text.

Response: We have deleted the sentence for the measurement of UV spectrum. However, we added the PDA data in the supporting information, because we could see the maximum of UV absorption by UPLC analysis.

  1. Page 8, line 217; There is no experimental description of the conversion of compound 2to MTPA ester. Also, there is no information on the spectral data of MTPA esters.

Response: We have added the data for the two MTPA esters of 2 in the ‘Materials and Methods’ section.

Round 2

Reviewer 1 Report

All the comments were well addressed. 

Author Response

Manuscript Number: marinedrugs-1317021

TITLE: New Cytotoxic Cytochalasans from A Plant-Associated Fungus Chaetomium globosum kz-19

Dear editor and reviewer:

We appreciate that you reviewed the manuscript very carefully and gave us insightful advices. We have rewritten and addressed the reviewer’s issues with blue highlights in the 2nd revised edition. 

Reviewer 2 Report

The authors revised manuscript as suggested by reviewers. However, there are several points to be further addressed. Therefore, the acceptance of the manuscript can be considered after further clarification.

- The authors selected two conformers described as 1a and 1b in lines 93-94 and Figure S9.  However, in the main text, 1a and 1b is enantiomeric pair line 94. It is impossible.

- 2a and 2b were included in Figure 2, but they are not addressed in main body of the manuscript.

- The ECD data value in the spectroscopic data in lines 233-246 are inconsistent with ECD spectra in the manuscript and supplementary materials.

Author Response

Manuscript Number: marinedrugs-1317021

TITLE: New Cytotoxic Cytochalasans from A Plant-Associated Fungus Chaetomium globosum kz-19

Dear editor and reviewer:

We appreciate that you reviewed the manuscript very carefully and gave us insightful advices. We have rewritten and addressed the reviewer’s issues with blue highlights in the 2nd revised edition. Please see the corresponding responses below.

Reviewer 2

  1. The authors selected two conformers described as 1a and 1b in lines 93-94 and Figure S9.  However, in the main text, 1a and 1b is enantiomeric pair line 94. It is impossible.

Response: Thank you for your comment.

We have corrected the word as ‘configurations’ in line 93-94. Additionally, in Figure S9, we revised the description of the conformers C1 and C2 as ‘C1 (92.55%)’ and ‘C2 (6.92%)’.

  1. 2a and 2b were included in Figure 2, but they are not addressed in main body of the manuscript.

Response: Thank you for your comment.

Line 131:We have revised the sentence as follows. ‘In addition, the ECD spectrum of 2 was determined and the Cotton effects were identical with the calculated curve of the enantiomer 2a (Figure 2c)…’

  1. The ECD data value in the spectroscopic data in lines 233-246 are inconsistent with ECD spectra in the manuscript and supplementary materials.

Response: Thank you for your comment.

Line 233-246:We have checked all the CD data of the four new compounds, and found that the data of compound 1 is wrong, which is different from the ECD result in Figure 2. We corrected the data as the following: CD (MeOH) 300 (Δε –0.97) nm, 270 (Δε +1.33), 226.5 (Δε –2.78), 211 (Δε +2.14) nm.

Reviewer 4 Report

This revised manuscript has been modified according to the reviewer’s comments. It is acceptable for publication.

Author Response

(The authors gave the same response as above.)
